
Geoscientific Model Development Discussions Open Access

# 1 The impact of resolving the Rossby radius at mid-latitudes

# 2 in the ocean: results from a high-resolution version of the

# 3 Met Office GC2 coupled model

**Helene T. Hewitt[1], Malcolm J. Roberts[1], Pat Hyder[1], Tim Graham[1], Jamie Rae[1],**
**Stephen E. Belcher[1], Romain Bourdallé-Badie[4], Dan Copsey[1], Andrew Coward[2],**
**Catherine Guiavarch[1], Chris Harris[1], Richard Hill[1], Joël J.-M. Hirschi[2], Gurvan**
**Madec[2,3], Matthew S. Mizielinski[1], Erica Neininger[1], Adrian L. New[2], Jean-**
**Christophe Rioual[1], Bablu Sinha[2], David Storkey[1], Ann Shelly[1], Livia Thorpe[1],**
**and Richard A. Wood[1]**
[1]{Met Office, Exeter, United Kingdom}
[2]{National Oceanography Centre, Southampton, United Kingdom}
[3]{IPSL, Paris, France}
[4]{Mercator Océan, Toulouse, France}
Correspondence to: H. T. Hewitt (helene.hewitt@metoffice.gov.uk)
**Abstract**
There is mounting evidence that resolving mesoscale eddies and boundary currents in the
surface ocean field can play an important role in air-sea interaction associated with vertical
and lateral transports of heat and salt. Here we describe the development of the Met Office
Global Coupled Model version 2 (GC2) with increased resolution relative to the standard
model: the ocean resolution is increased from 1/4° to 1/12° (28km to 9km at the Equator), the
atmosphere resolution increased from 60km (N216) to 25km (N512) and the coupling
frequency increased from 3-hourly to hourly. The technical developments that were required
to build a version of the model at higher resolution are described as well as results from a 20
year simulation. The results demonstrate the key role played by the enhanced resolution of the
ocean model: reduced Sea Surface Temperature biases, improved ocean heat transports,
deeper and stronger overturning circulation and a stronger Antarctic Circumpolar Current.
Our results suggest that the improvements seen here require high resolution in both





atmosphere and ocean components as well as high frequency coupling. These results add to
the body of evidence suggesting that ocean resolution is an important consideration when
developing coupled models for weather and climate applications.
**1   Introduction**
On the scale of the Rossby radius, the ocean is rich with mesoscale eddies (Chelton et al.,
2011) and oceanic fronts.   There is mounting evidence from satellite observations that
mesoscale features in the Sea Surface Temperature (SST) field can drive comparable
variations in atmospheric winds and surface fluxes (Chelton and Xie, 2010; Frenger et al.,
2015). While at the basin scale, observed correlations between SST and surface winds are
negatively correlated, indicating that the atmosphere is driving the ocean, in frontal regions
with high mesoscale activity, such as those associated with Western boundary currents, SST
and surface winds are positively correlated, implying that the ocean is driving the atmosphere
(Bryan et al., 2010). While the primary response to SST takes place in the atmospheric
boundary layer (Chelton and Xie, 2010), there is also evidence that divergence of surface
winds may give rise to vertical motions which may penetrate high into the troposphere
affecting storm tracks and clouds (e.g., Minobe et al., 2008; Sheldon and Czaja, 2014). Of
particular note is the intense rain band in the North Atlantic that follows the path of the Gulf
Stream/North Atlantic Current.
The recent CMIP5 ocean models have a horizontal resolution of between 1° and 1/4°.
However, with a resolution of 28km at the Equator down to 6km in the Canadian archipelago
(due to the tripolar grid), even 1/4° remains insufficient to resolve mesoscale eddies which
have a typical scale of 50km in the deep ocean at mid-latitudes (Hallberg, 2013). Several
climate modelling groups have now built global coupled models with an "eddy resolving"
component (e.g., McClean et al., 2011; Bryan et al., 2010; Delworth et al., 2012; Small et al.,
2014; Griffies et al., 2015). In this paper, we describe results from coupling the 1/12° ocean
model (ORCA12) produced by the Drakkar group (Marzocchi et al., 2015; Deshayes et al.,
2013; Treguier et al., 2012) to a 25 km (N512) resolution version of the Met Office Unified
Model (MetUM) atmosphere. This is the first version of the HadGEM3/GC series (Hewitt et
al., 2011; Williams et al., 2015) to resolve the Rossby radius in the ocean at mid-latitudes
(with a resolution of 9km at the Equator down to 2km in the Canadian archipelago) and the
first coupled experiment with the NEMO ORCA12 ocean configuration.



Evidence from forced ocean simulations demonstrates that resolution enables a more realistic
representation of both eddy kinetic energy (Hurlburt et al., 2009; Griffies et al., 2015), narrow
boundary currents (e.g., Marzocchi et al., 2015) and representation of complex topography, in
particular the sills which connect ocean basins (e.g., improved overflows in the VIKING
model at 1/20° resolution; Behrens et al., 2013). In this paper we investigate how ocean
resolution drives large-scale changes not only in the ocean but also in the climate system.
Changes in the ocean circulation could be important both for present and future climate; for
example, in an ocean-only model with a simple domain, Zhang and Vallis (2013) have shown
that the changes in mean circulation due to eddy-resolving resolution can affect the net ocean
heat uptake under global warming scenarios.
In this paper, the model is described in section 2. Our results (section 3) describe the relative
impact of the three changes to the model; ocean resolution, atmosphere resolution and
coupling frequency. Finally in section 4 we summarise and discuss the results.

## 2    Model description

The development of the high resolution coupled climate model is based on the Met Office
Global Coupled model version 2 (GC2; Williams et al., 2015). GC2 is comprised of the Met
Office Unified Model (MetUM; GA6) atmosphere, the JULES land surface model (Best et al.,
2011; GL6), the NEMO ocean model (Madec, 2014; GO5: Megann et al., 2014) and the Los
Alamos CICE sea-ice model (Hunke et al., 2010; GSI6: Rae et al., 2015). The standard
configuration for GC2 has a 60km resolution atmosphere coupled to 1/4° (28km at the
Equator reducing polewards) ocean (N216-ORCA025) with coupling between the
components (as described in Hewitt et al., 2011) every three hours. GA6 has 85 vertical levels
while GO5 has 75 vertical levels with 1m resolution in the top 10m of the ocean (Megann et
al., 2014). Although vertical resolution is not explored here, we include details of the vertical
levels in appendix A.
In addition to GC2, this paper describes three modified versions of GC2 with increased
atmosphere resolution, increased coupling frequency and increased ocean resolution. The
different model experiments are described below and summarised in Table 1.
GC2 has been run with a high 25km (N512) atmosphere resolution and the standard
(ORCA025) resolution ocean and we will refer to this as GC2-N512. The scientific



differences between N216 and N512 are minimal, as described in Walters et al. (in prep), and
are principally associated with the time step (modified from 15min to 10min) and the
resolution of the external boundary conditions such as the orography.
To facilitate direct scientific comparison with the 1/12° ORCA12 (9km at the Equator
reducing polewards) configuration of NEMO, which was developed using NEMO v3.5 rather
than 3.4 (Marzocchi et al., 2015), a modified configuration of GC2, referred to here for
convenience as GC2.1 was developed. The key scientific and technical changes made to
GC2.1 are:
• an increase in the coupling frequency from 3-hourly to hourly
• an upgrade to the non-linear free surface scheme rather than the linear free surface
• a small reduction in the timestep from 1350s to 1200s (to accommodate hourly
coupling)
• small changes associated with river outflows; outflows prescribed over 15m rather
than 10m with an enhanced vertical mixing in the outflow region of $1\times10^{-3}\,\mathrm{m}^2\mathrm{s}^{-1}$ rather
than $2\times10^{-3}\,\mathrm{m}^2\mathrm{s}^{-1}$
• an upgrade of the sea ice model from CICE4 to CICE5 (Hunke et al., 2015). This
upgrade was for technical reasons and the science of the sea ice configuration remains
unchanged.
To assess the impact of ocean resolution, a traceable GC2.1 configuration with ORCA12 was
then built (further technical details and model performance issues are discussed in appendix
B). We chose to increase the atmosphere resolution to N512 in order to maintain a similar
aspect ratio of atmosphere to ocean grids. We will refer to this configuration as GC2.1-
N512O12 (i.e., increased atmosphere and ocean resolution).
The differences between ORCA025 and ORCA12 in GC2.1 are:
• a reduction in the time step from 1200s to 240s
• a reduction in the isoneutral tracer diffusion from 300 $\mathrm{m}^2\mathrm{s}^{-1}$ to 125 $\mathrm{m}^2\mathrm{s}^{-1}$
• a reduction in the bilaplacian viscosity from $-1.5\times10^{11}\,\mathrm{m}^2\mathrm{s}^{-1}$ to $-1.25\times10^{10}\,\mathrm{m}^2\mathrm{s}^{-1}$
We note here that the parameter settings in GC2.1-N512O12 have not been tuned for the
coupled model; the model was run using the majority of parameter settings from the forced
ocean-only ORCA12 runs of Marzocchi et al. (2015).



GC2.1-N512O12 was found to be very sensitive to features that had not proved to be a
problem in previous ocean-only integrations (e.g., Marzocchi et al., 2015). For example, the
model became unstable on the east coast of the UK every 6-12 months of simulation due to
extreme values in the velocity field, likely due to the lack of tides in the model which are very
important in this region. The model was restarted from these failures with a small random
perturbation to the atmospheric theta field in a similar way to treatment of "grid-point
instabilities" previously seen in atmosphere models (e.g., Mizielinski et al 2014). The
underlying problem with this unstable ocean point will be addressed in future developments
of the ORCA12 configuration.
The GC2 and GC2.1 experiments were run for 20 years with fixed atmospheric radiative
forcing representative of the present day (with greenhouse gas and aerosol values for the year
2000). All experiments were initialised in the following way:
• atmosphere: N216 and N512 both from September year 18 of the model state of a

14       previous N512 GA6 (Walters et al., in prep) forced atmosphere integration with

15       forcing representative of the year 2000, so that the land surface properties are at quasi-

16       equilibrium;

• ocean:  temperature and salinity from the EN3 observational dataset (Ingleby and

18       Huddleston, 2007) 2004-8  September average with velocities initialised to zero;

• sea ice: 20 year September mean from a HadGEM1 (Johns et al., 2006) experiment

20       representative of a period centred on 1978.

• These latter two are the standard method for initialisation of "present day" coupled

22       simulations at the Met Office.

The choice of the most appropriate ratio between ocean and atmosphere resolution remains an
open research question worthy of further study. Short (two year) integrations using both
higher and lower atmosphere resolutions coupled to ORCA12 were completed, although due
to the short length of the integrations, they are not analysed here. In particular, a configuration
using an N768 (17km) atmosphere led to a marked increase in the frequency of model
instabilities (5-6 per year).



## 3 Impact of model resolution on surface properties, heat transport and ocean circulation

The results shown in this section derive from 20 year simulations of the four experiments described in table 1, initialised and forced in an identical way.

### a. Surface Properties

The pattern of large-scale biases in SST fields in Hadley Centre coupled climate models have remained largely unchanged since the models first ran without flux correction (e.g., Gordon et al., 2000); the large-scale biases exhibit warming in the Southern Ocean, cooling in the North Pacific and North Atlantic and warming in upwelling/stratocumulus regions off the western coasts of South America and Africa. Many of these biases are also very common in other models (e.g. Small et al., 2014).

The time-series of the global mean Top of Atmosphere (TOA) radiation imbalance in the four models (Figure 1a) shows that the experiments with high (N512) atmosphere resolution have TOAs that are generally higher at the start of the experiments. However after 20 years all the experiments are starting to converge to a similar net TOA, as the shortwave and long-wave components adjust. Although the TOA-SST relationship is poorly defined (since the TOA imbalance is related to the rate of change of net ocean heat content; Palmer and McNeall, 2014), the integrated effect of the higher net TOA in the N512 experiments can be seen in the timeseries of the global mean SST (Figure 1b) with GC2-N512 and GC2.1-N512O12 having higher global mean SSTs.

In spite of the differences in global mean SST, major changes to the pattern and magnitude of SST biases are only seen with both high atmosphere and ocean resolution (Figure 2). In GC2.1 N512-ORCA12, the large-scale underlying SST biases are reduced relative to GC2 and GC2.1 (Figure 3): the warm bias in the Southern Ocean; cold bias in North Atlantic and North Pacific and warm biases in stratocumulus regions. Similar reductions in SST biases with high atmosphere and ocean resolution were also seen in Small et al. (2015). The increase in ocean resolution is key to this improvement: when only atmosphere resolution is increased (compare Figures 2a and b), there is only a small reduction in the warm bias associated with stratocumulus regions (west of South America and Africa), while increased coupling frequency (compare Figures 2a and c) shows only minor changes in SST biases.



In GC2 there is a cold bias in the North Atlantic subpolar gyre (SPG), Greenland-Iceland-
Norwegian (GIN) Seas and the Arctic. GC2.1-N512O12 shows a warming of several degrees
in the SPG and GIN seas relative to GC2 (see reduced cold bias in Figure 2d) and a very large
warming in the Central Arctic. The warming in the Central Arctic is associated with a
warming in the subpolar gyre, enhanced northward heat transport into the Arctic and melting
back of the sea ice edge in the Arctic (see below).
Resolution appears to have less of an impact on Sea Surface Salinity (SSS; Figure 4).
Nevertheless, there are reductions in high salinity biases in the Indian Ocean and the Pacific
(in particular, in the salinity maximum in the subtropical gyre of the South Pacific) as well as
reductions in the Arctic biases (although these are very sensitive to the distribution of sea ice).
*b.  Sea ice*
The changes to the SST also affect sea ice distribution in both hemispheres. The seasonal
cycle of ice extent in the Arctic (Figure 5a) shows that the warm SSTs in GC2.1-N512O12 at
high Northern latitudes reduce the ice extent throughout the year. The March ice
concentrations in the Arctic (Figure 6) clearly demonstrate that the impact on the sea ice is
concentrated in the GIN seas with the sea ice edge in GC2.1-N512O12 much further north
than seen in GC2 with the edge being north of Spitzbergen and into the Barents Sea.
In comparison, the reduction in the warm bias in the Southern hemisphere leads to only
modest increases in the total sea ice extent (Figure 5b); the overall warming bias associated
with the lack of super-cooled liquid clouds (Bodas-Salcedo et al., 2014; Bodas-Salcedo et al.,
in press) still dominates the melting of sea ice. The small increase in sea ice extent is very
inhomogeneous; indeed, some regions in the Southern Ocean such as the Weddell Sea
actually show reductions in sea ice extent in GC2.1 N512-ORCA12 (Figure 6). The reduction
in the Weddell Sea is associated with a polynya in that region (see below).
*c.  Sub-surface ocean drifts*
Conservation of heat within the climate system implies that the net heat uptake by the ocean
should nearly balance the net radiative imbalance at the TOA. GC2.1-N512O12 has the
highest TOA imbalance of the four models (Table 2) and therefore will have the greatest net
heat uptake. Both models with increased atmosphere resolution (GC2-N512 and GC2.1-





N512O12) have a higher TOA imbalance than the models with lower atmosphere resolution
(GC2 and GC2.1).
The global temperature profiles (Figure 7a) show that GC2-N512 and GC2.1-N512O12 do
indeed have greater increases in temperature as a function of depth than either of the low
resolution models (GC2 and GC2.1), which is consistent with the higher TOA imbalance. The
main difference between GC2-N512 and GC2.1-N512O12 is that the increase in heat uptake
extends deeper in GC2.1-N512O12. This difference is also apparent in the global mean SST
anomaly (Table 2); the SST anomaly for years 11-20 in GC2.1-N512O12 is 0.44 K compared
with 0.60 K in GC2-N512, while the TOA imbalance is 2.02 $W/m^2$ and 1.79 $W/m^2$
respectively. This shows that the ORCA12 version of the model is able to transport heat to
depth more effectively.
The distribution of the subsurface temperature changes varies depending on the latitudinal
range. South of 30°S (Figure 7b), near surface warming is reduced in GC2.1-N512O12
relative to the other models. In the Tropics (30°S-30°N; Figure 7c), GC2.1-N512O12 shows
increased warming shallower than 500m relative to the low resolution models but reduced
relative to GC2-N512. The Tropics also show increased warming at depth in GC2.1-
N512O12. The largest increase in near surface temperatures in GC2.1-N512O12 relative to
the other models occurs north of 30°N (Figure 7d) with the surface warming displacing a cold
bias to deeper in the water column. The warming is particularly concentrated north of 65°N
(Figure 7e) where it has previously been shown that Arctic sea ice melts back.
Drifts in sub-surface salinity show that GC2.1-N512O12 generally has larger salinity drifts
between 500 and 1000m (Figure 8a) which is largely associated with the region south of 30°S
(Figure 8b). In the northern hemisphere, drifts in salinity between 1000 and 2000m are also
more pronounced in GC2.1-N512O12 than the other models (Figure 8d). In contrast, large
fresh biases north of 65°N in most of the models is much reduced in GC2.1-N512O12 (Figure
8e). Understanding salinity drifts and their relationship to freshwater forcing is complex (eg,
Pardaens et al. 2003) and this aspect of the model performance will require further
investigation.





*d.  Mixed layer depths*
In general over the open oceans, the mixed layer depths (Figure 6) are very similar across the
different models and it is in the deep water formation regions where we see inter-hemispheric
changes. Winter mixed layers in the Northern hemispheres in GC2.1-N512O12 show a
reduction in the North Atlantic subpolar gyre. Most notably, in GC2.1-N512O12 deep mixed
layers are less extensive south of Greenland than in GC2 and are confined to the centre of the
Labrador Sea. Similar changes in Labrador Sea deep convection have been seen in sensitivity
experiments when overflow properties are improved (Graham et al., in prep.). The deeper
mixed layers in the Arctic in GC2.1-N512O12 are consistent with warmer SSTs and reduced
sea ice extent in that region exposing open water.
The similarity of the mixed layer depths across the Southern Ocean demonstrate that it is not
changes to the mixed layer depths that lead to a reduction in the Southern Ocean warm bias.
As mentioned in the previous section, in the Weddell Sea, GC2.1-N512O12 has very deep
mixed layers linked to a polynya, which explains the lack of increase of sea ice extent in that
region (Figure 6). Deeper winter mixed layers in GC2.1-N512O12 are also evident through
the mid-latitudes in the formation zones for Sub-Antarctic Mode Waters and Antarctic
Intermediate Waters. These could be due to the reduced warm bias (cooler SSTs) in these
regions (Figure 2).
*e.  Ocean Circulation*
The improvements seen in the large-scale SST biases with high atmosphere and ocean
resolution (Figure 3) represent an overall improvement in the model simulation with warming
in the Northern hemisphere and cooling in the Southern hemisphere. This pattern is
reminiscent of inter-hemispheric modes that occur as a result of changes in the large-scale
thermohaline circulation (Vellinga and Wu, 2004). The meridional overturning in our
simulations changes only in the GC2.1-N512O12, with an increase of O(3 Sv) (Table 2) both
in the North and South Atlantic, and is therefore attributed to the increased ocean resolution.
The changes in the meridional overturning circulation (Figure 9) are dominated by changes in
the cell associated with North Atlantic Deep Water (NADW) with changes extending into the
Southern hemisphere.





At the northern end of the NADW cell, we see increases in the volume flux of dense
overflows between the GIN Seas and the Atlantic (Table 2) that are consistent with the
NADW cell being strengthened both by the GIN sea sources and better representation of sills.
The volume flux of overflow waters across Denmark Straits generally reduces fairly rapidly
in ORCA025 runs (Figure 10a) but in GC2.1-N512O12 the overflow remains closer to the
observed value of 2.9 - 3.7 Sv (Dickson and Brown, 1994; Macrander et al., 2005). This
appears to also contribute to a deeper (as well as stronger) NADW outflow in this model and
is almost certainly associated with the increased resolution of the topography in the region of
the overflows.
The Antarctic Circumpolar Current (ACC) usually drifts in the ORCA025 GC models from
an initial value of approximately 150 Sv to below 100 Sv (Figure 10b). Increased ocean
resolution counteracts that, with the ACC stabilising close to 130 Sv in this 20 year
experiment. This value is close to the observations that suggest an ACC transport of $137 \pm 8$
Sv (Cunningham et al., 2003)). The increase in the transport in the ACC can be explained by
changes in the density field; the meridional density gradients across the ACC (not shown) are
increased in GC2.1-N512O12 (with steeper isopycnals) than in GC2 which is consistent with
increased southward flow, and stronger upwelling, of NADW to the north of the ACC and
increased convection to the south of the ACC in the Weddell Sea. The Southern Ocean winds
(not shown) respond differently across the four simulations (including a small increase in
GC2.1-N512O12 and a decrease in GC2.1) and investigating these changes, how they relate
to the model internal variability and their impact on the simulation will be a topic of future
research.
*f. Heat transport*
As described in Gordon et al. (2000), drifts in volume averaged ocean temperature can be
related to discrepancies between the actual heat transports by the ocean and the heat transport
implied by the surface fluxes, i.e.
$$\frac{\partial \rho c_p <\theta>}{\partial t} + \oint \rho c_p (\bar{v}\bar{\theta} + \overline{v'\theta'})dS + \oint \rho c_p A_{iso} \nabla_\rho \theta \, dS = \int F \, dA, \qquad (1)$$
where $<\theta>$ is the volume integrated temperature, $\bar{v}\bar{\theta}$ and $\overline{v'\theta'}$ are the time mean and time
varying components of the ocean meridional heat transports, $\rho c_p$ is density multiplied by
specific heat capacity, $A_{iso}$ is the isopycnal diffusion, $\nabla_\rho \theta$ is the isoneutral gradients of



temperature and F is the surface heat flux. For our purposes here, we make the assumption
that the isoneutral fluxes are generally smaller than the other terms (dianeutral diffusive
fluxes are very small when integrated over full depth).
Figure 11a shows the global northward heat transport in all four simulations. There are some
changes in the northern hemisphere in the GC2.1 simulation with the change to hourly
coupling, while changes in the southern hemisphere are only seen in GC2.1-N512O12
suggesting that these changes are driven by the increase in ocean resolution. The reduction in
southward heat transport in GC2.1-N512O12 centred at 45°S is highly unusual; although the
change does not lie outside interannual variability, a change of this magnitude in the multi-
year mean heat transport has not been seen in any other development runs of the GC series.
The modelled changes in the heat transports suggest that ocean processes are important in this
region, which is particularly relevant given the uncertainty in surface heat fluxes in the
Southern Ocean (Cerovecki et al., 2011). The change in total heat transport comes primarily
from the time mean heat transport (not shown). This suggests that changes in resolution have
led to a change in either the mean circulation or the temperature profile (as opposed to a
change in the time varying heat transport, which would imply a direct role of the mesoscale
eddies). As seen in previous sections, GC2.1-N512O12 shows changes in both the circulation
and the temperature profiles. The decreased southward heat transport in the Southern Ocean
of GC2.1-N512O12 could – at least partly - explain the reduced warm bias.
By comparing actual ocean heat transports with those implied by surface fluxes (i.e., the
second term of the left-hand side of Eqn. 1 with the right-hand side of Eqn. 1), this gives an
indication of the volume averaged drift in temperature (first term on the left-hand side of Eqn.
1). The implied ocean heat transport is calculated by subtracting the globally averaged
imbalance from the surface fluxes before integrating zonally and meridionally. Globally
(Figure 11a) GC2.1-N512O12 can be seen to be as close to local balance as any of the other
models, suggesting that the net drifts will be of a similar magnitude (in agreement with Figure

27 5).

Ocean resolution is the driving factor in a 0.2PW increase in the northward heat transport in
the Atlantic; the modelled heat transports in GC2.1-N512O12 are generally within the error
bars of the observations (Ganachaud and Wunsch, 2003; Figure 11b) in contrast to the other
models with the lower resolution ocean component. The change in heat transport is linked to
an increase in the overturning circulation (previous section), which is unsurprising given the



dominant role of the meridional overturning circulation in the Atlantic heat transport (Hall
and Bryden, 1982).
**4    Summary and Discussion**
In this paper we have shown results from a coupled climate model with an eddy resolving
(1/12°) ocean component coupled to a high resolution (25 km) atmosphere component. When
the SST bias from this climate simulation is compared to that from the Met Office standard
resolution climate model, with eddy permitting (1/4°) ocean component and 60km atmosphere
component, it is apparent that major SST biases in the Southern Ocean and North Atlantic and
North Pacific have been reduced. Comparable experiments increasing only the atmosphere
resolution or the coupling frequency, demonstrate that increased ocean resolution is the key
driver for this change.
At the enhanced ocean resolution, the ocean circulation leads to increased poleward ocean
heat transport in the Northern hemisphere and reduced poleward ocean heat transport in the
Southern hemisphere. The change in the northward heat transport is driven at least in part by
an enhanced NADW cell which also contributes to maintaining the ACC front. The ACC
front is maintained in spite of the expectation that improved representation of eddies in the
Southern Ocean could lead to slumping of the front, this is at least in part associated with
enhanced windstresses at high resolution. Changes in the global heat transports produce a
shift in the large-scale biases, cooling the Southern Ocean and warming the North Atlantic
and North Pacific. We have shown that heat penetrates deeper in our 1/12° model; Griffies et
al. (2015) have demonstrated that mesoscale eddies transport heat upwards so it is likely that
the increased transport of heat to depth is achieved by the time-mean as seen in transient
experiments such as Banks and Gregory (2006). Future work will be focused on
understanding the relative roles of resolving overflow topography (Behrens, 2013), eddy
processes within the ocean including compensation and saturation (e.g., Munday et al., 2013)
and air-sea interaction on the eddy scale (Roberts et al., in prep.) in driving the large-scale
changes.
Relative to the recent high resolution results of Small et al. (2014) and Griffies et al. (2015),
our results emphasise the importance of increasing both atmosphere and ocean resolution.
Griffies et al. (2015) show smaller reductions in SST biases when moving from 1/4° to 1/10°
resolution presumably related to keeping the atmosphere resolution unchanged. Enhanced



coupling frequency along with enhanced vertical resolution near the air-sea interface both in
the ocean (Megann et al., 2014) and atmosphere (Walters et al., in prep) is one feature of our
model setup that is missing in Small et al. (2014). These aspects of the model setup may be
especially important in regions of strong air-sea interaction including the stratocumulus
regions where we see large improvements in the GC2.1-N512O12 simulation. Overall, the
improvements seen in this paper required a combination of high resolution in both atmosphere
and ocean components as well as high frequency coupling.
As described in the previous section, one of the changes to the ocean model at higher
resolution was a reduction in the isoneutral diffusion. Pradal and Gnanadesikan (2014) show
that a reduction in the isoneutral diffusion from 800 $m^2s^{-1}$ to 400 $m^2s^{-1}$ in a coarse resolution
climate model is associated with cooling of order 1°C at high latitudes after 500 years. While
the results here may exhibit some consistency with those of Pradal and Gnanadesikan (2014)
in the Southern Ocean, the change in isopycnal diffusion is believed to be a secondary effect
due to the fact that we are seeing a comparable or larger change in SST in a short 20 year run.
One caveat of these results is that the parallel simulations lasted only 20 years. However, the
close agreement between implied and actual meridional heat transports, suggests that the
models are close to quasi-equilibrium. Additionally, the broad similarity of the results
presented here compared with those of Small et al. (2014) from over 100 years of simulation
suggest that the results are reasonably robust. In terms of model drift, climate models
typically have a fast adjustment within the first five years (Sanchez-Gomez et al., 2016).
Large adjustments over the first 20 years are generally followed by a multi-centennial drift
towards equilibrium between ocean properties and the net TOA flux (Banks et al., 2007).
Longer simulations and further analyses will enable the robustness of the results presented
here (including wind-SST feedbacks) to be more fully understood.
In the results here, the 1/12° ocean model, which has a resolution of approximately 7 km at
mid-latitudes, is coupled to an N512 atmosphere model, which has a resolution of 25 km. An
atmosphere:ocean ratio of 4:1 may be too high for the atmosphere to fully capture the details
of the ocean mesoscale. Future work will investigate the impact of coupling to even higher
resolution atmosphere models to investigate the role of the atmosphere:ocean ratio.
As we move towards seamless coupled prediction, using coupled models for prediction on
timescales from days to centuries, the results presented here are highly relevant to prediction
up to decadal timescales where data assimilation is employed. A coupled model that more





faithfully produces the current state of the ocean will rely less on data assimilation for
correcting large-scale biases and be more able to include the representation of spatial
anomalies that control the large-scale variability. While there are many regions where
subsurface drifts are improved at this resolution, reducing the drifts seen in mid-depth salinity
will be important.
A key question for these timescales is whether employing enhanced resolution will address
the known problem of low signal-to-noise ratios (Eade et al., 2014) that has led to the need for
large ensembles for seasonal to decadal forecasting in lower resolution systems. Future work
to understand the drivers of large-scale bias reduction will support targeted experiments to
address the relative roles of resolution and ensemble size at these timescales. That said, ocean
resolution is clearly not going to solve all the issues in climate models; atmosphere errors
often dominate surface biases and, even at high resolution, ocean models need improved
representation of sub-gridscale processes.
**Code availability**
The MetUM is available for use under licence. A number of research organizations and
national meteorological services use the MetUM in collaboration with the Met Office to
undertake basic atmospheric process research, produce forecasts, develop the MetUM code
and build and evaluate Earth system models. For further information on how to apply for a
licence see http://www.metoffice.gov.uk/research/collaboration/um-collaboration. JULES is
available under licence free of charge. For further information on how to gain permission to
use JULES for research purposes see https://jules.jchmr.org/software-and-documentation. The
model code for NEMO v3.4 and v3.5 is available from the NEMO website (www.nemo-
ocean.eu). On registering, individuals can access the code using the open source subversion
software (http://subversion.apache.org/). The model code for CICE is freely available
(http://oceans11.lanl.gov/trac/CICE/wiki/SourceCode) from the United States Los Alamos
National Laboratory. In order to implement the scientific configuration of GC2/GC2.1 and to
allow the components to work together, a number of branches (code changes) are applied to
the above codes. Please contact the authors for more information on these branches and how
to obtain them.





**Appendix A: Model vertical levels**
The sensitivity to vertical resolution is not explored in this paper. However, a reduced
description of the vertical levels in GA6 (Table A1) and GO5 (Table A2) are included to
allow comparison with other models. For the full vertical levels, see Walters et al. (in prep.)
and Megann et al. (2014), respectively.

| Level | Rho_height (m) |
|-------|----------------|
| 1 | 10.00 |
| 10 | 730.00 |
| 20 | 2796.67 |
| 30 | 6196.67 |
| 40 | 10930.12 |
| 50 | 17012.40 |
| 60 | 24710.70 |
| 70 | 35927.89 |
| 80 | 58978.35 |
| 85 | 82050.01 |

Table A1: Reduced list of level in GA6 which has 85 vertical levels





| Level | Depth (m) | Thickness (m) |
|-------|-----------|---------------|
| 1 | 0.51 | 1.02 |
| 10 | 13.99 | 2.37 |
| 20 | 61.11 | 7.58 |
| 30 | 180.55 | 18.27 |
| 40 | 508.64 | 53.76 |
| 50 | 1387.38 | 125.29 |
| 60 | 2955.57 | 181.33 |
| 65 | 3897.98 | 194.29 |
| 70 | 4888.07 | 200.97 |
| 75 | 5902.06 | 204.23 |

2     Table A2: Reduced list of levels and layer thicknesses in GO5 which has 75 vertical levels





**Appendix B: Model performance and technical aspects**
The GC2.1 configuration was the first in which several further technical components of the
coupled system were considered essential to make the simulation manageable. The coupler
was upgraded from OASIS3 to OASIS3-MCT (Valcke et al, 2015) in order to improve
parallelisation of the coupling, particularly given the increased coupling frequency.
ORCA025 files are typically written as one file per processor by standard GC2 configurations
and combined into a single file prior to analysis as a post processing step. However, as HPC
parallel file systems are generally tuned for high bandwidth on large files and as GC2.1-
N512O12 configurations allocate 50 of the 80 nodes used by the full coupled system to the
ocean, this led to performance and functional issues when running on 1600 or more cores.
The NEMO XIOS diagnostic server (Madec, 2014) provides an asynchronous IO server
capability that allows the diagnostic files to be output as fewer larger files (although the
restart files are still written as one file per processor). Its introduction in the model allowed us
to overcome the limitations of the file system.
Land suppression was used for the NEMO and CICE models, so that processors are only
assigned to regions with active ocean points. This leads to a significant gain in core count,
although it meant that the automated large-scale diagnostics usually produced by NEMO
(zonal mean heat transports, meridional overturning) could not be generated.
Data volumes from this experiment were particularly large due to the output of additional
hourly and 3-hourly fluxes in order to examine the coupling processes in more detail. Each
month of model output comprised: ocean monthly mean files (netCDF) of 87GB together
with 6GB of daily files, sea-ice output (netCDF) of 57GB per month (with an additional
48GB of hourly output), and atmosphere output (PP format) of 100 GB per month. In total,
the 20 years of simulation produced 85 TB of data.
Little optimisation of the model was attempted since GC2.1 is not intended to be supported in
the  long-term. Its successor, GC3, will be used for CMIP6. The GC2.1-N512O12 model used
80 full nodes (each of 32 cores) of an IBM Power 7 HPC, of which 55 were allocated to the
ocean/sea ice component (including 5 for the IO servers) and 25 for the atmosphere/land
component. The model throughput was 4 months per wall-clock day.
For previous model resolutions, the SCRIP utility (Jones, 1998) was used to generate the
conservative remapping files used to regrid coupling data between the ocean and atmosphere



grids (for temperature and fluxes), with bilinear interpolation used for the winds and surface
currents. However, due to the size of the high resolution grids used here, and the serial nature
of SCRIP, a different method was required.  ESMF (ESMF, 2014; a package of parallelised
tools that use the same input grid descriptions as SCRIP, but can be run in parallel) was
therefore used to generate the remapping weights.
**Acknowledgements**
This work was primarily supported by the Joint DECC/Defra Met Office Hadley Centre
Climate Programme (GA01101). Part of the work was undertaken with National Capability
funding from NERC for ocean modelling. We acknowledge use of the MONSooN system, a
collaborative facility supplied under the Met Office-NERC Joint Weather and Climate
Research Programme (JWCRP). Met Office authors were supported by the joint UK
DECC/DEFRA Met Office Hadley Centre Climate Programme (GA01101).  MR
acknowledges support from the EU FP7 IS-ENES2 project for work on ESMF and regridding
tools. We acknowledge the considerable effort on development and evaluation of ORCA12 by
the DRAKKAR community. HH thanks IH.

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



Table 1. Coupled models used in this paper

| Model | Horizontal Resolution | Coupling frequency |
|---|---|---|
| GC2 (Williams et al., 2015) | N216-ORCA025 | 3-hourly |
| GC2-N512 | N512-ORCA025 | 3-hourly |
| GC2.1 (this paper) | N216-ORCA025 | 1-hourly |
| GC2.1-N512O12 | N512-ORCA12 | 1-hourly |

Table 2. Key metrics from years 11-20 of the experiments and observations. TOA
observations from CERES/EBAF for years 2000-2010. Global mean SST error (compared to
Reynolds OI). Overflows are calculated as southward flow across the Greenland-Iceland-
Scotland ridge below density of 27.8 kg m$^{-3}$ and have standard deviation shown in brackets.

| Model | Net TOA (W/m$^2$) | Global mean SST error (K) | Maximum overturning at 30°S (Sv) | Maximum overturning at 24°N (Sv) | Net transport from overflows (Sv) |
|---|---|---|---|---|---|
| Observations | 0.85 | | | | |
| GC2 | 1.61 | 0.25 | 13.7 | 14.6 | 4.0 (0.24) |
| GC2-N512 | 1.79 | 0.60 | 14.3 | 14.9 | 3.9 (0.28) |
| GC2.1 | 1.64 | 0.29 | 14.3 | 16.4 | 4.7 (0.26) |
| GC2.1-N512O12 | 2.02 | 0.44 | 17.5 | 17.7 | 5.9 (0.42) |



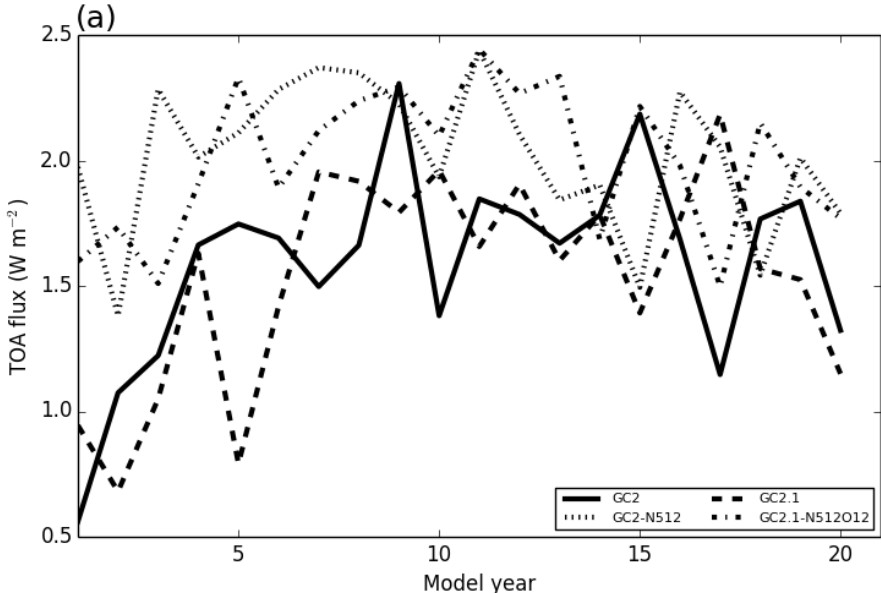

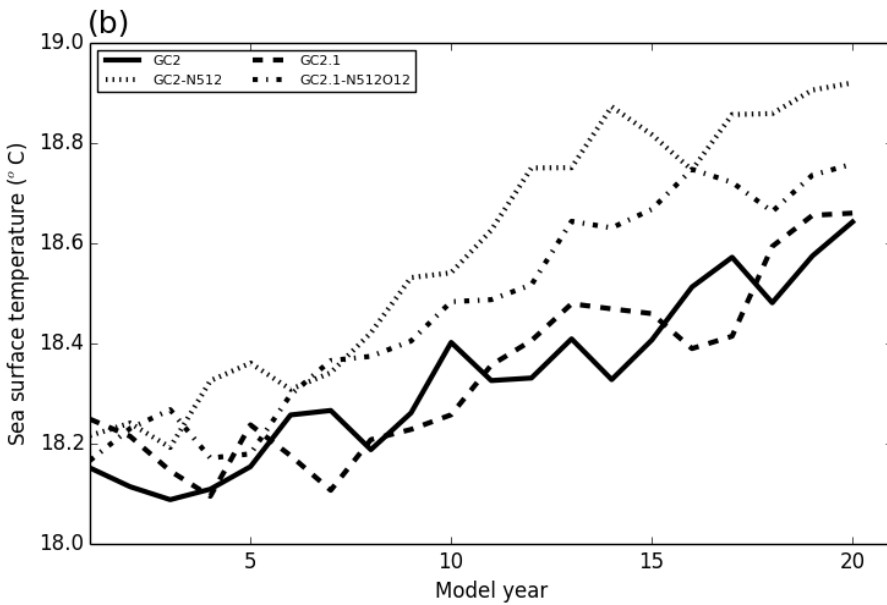

5    Figure 1: Timeseries of a) net TOA and b) global mean SST from GC2, GC2-N512, GC2.1

6    and GC2.1-N512O12.



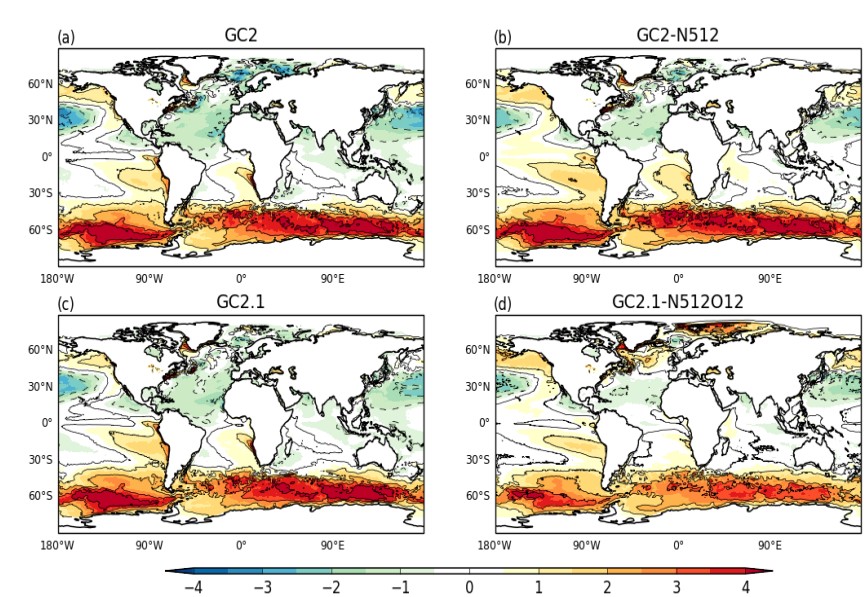

4 Figure 2: Differences between modelled SST from years 11-20 and observed SST from
5 HadISST (°C) for a) GC2, b) GC2-N512, c) GC2.1 and d) GC2.1-N512O12.





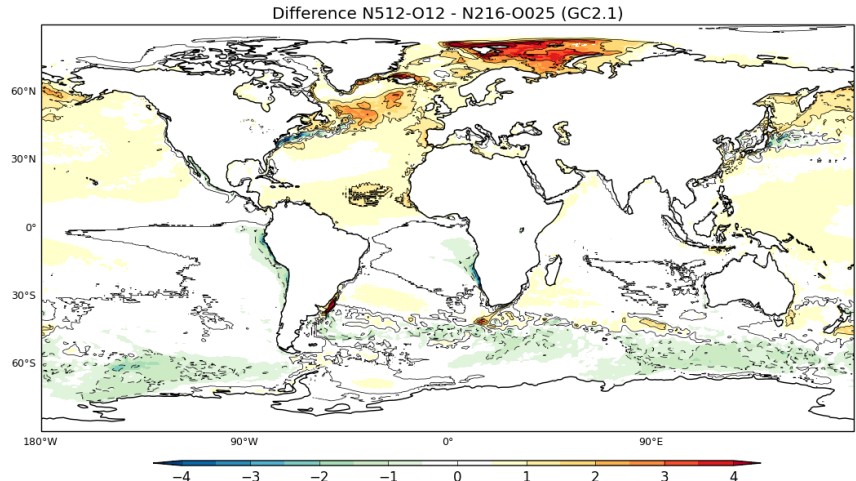

3       Figure 3: SST difference (°C) for years 11-20 between GC2.1-N512O12 and GC2.1



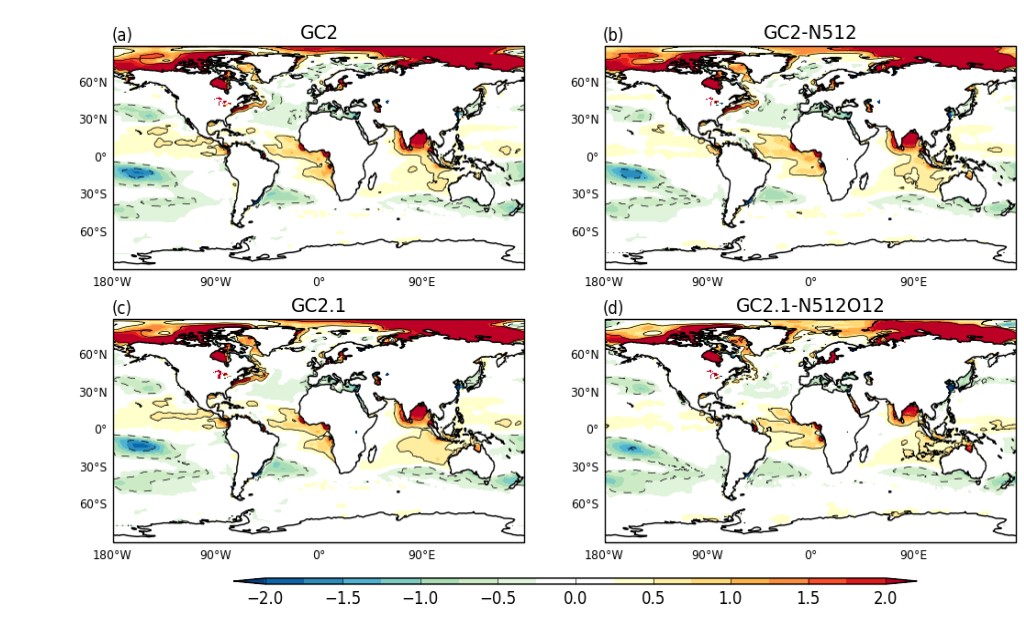

Figure 4: Differences between modelled SSS from years 11-20 and observed SSS from EN4

(psu) for a) GC2, b) GC2-N512, c) GC2.1 and d) GC2.1-N512O12.



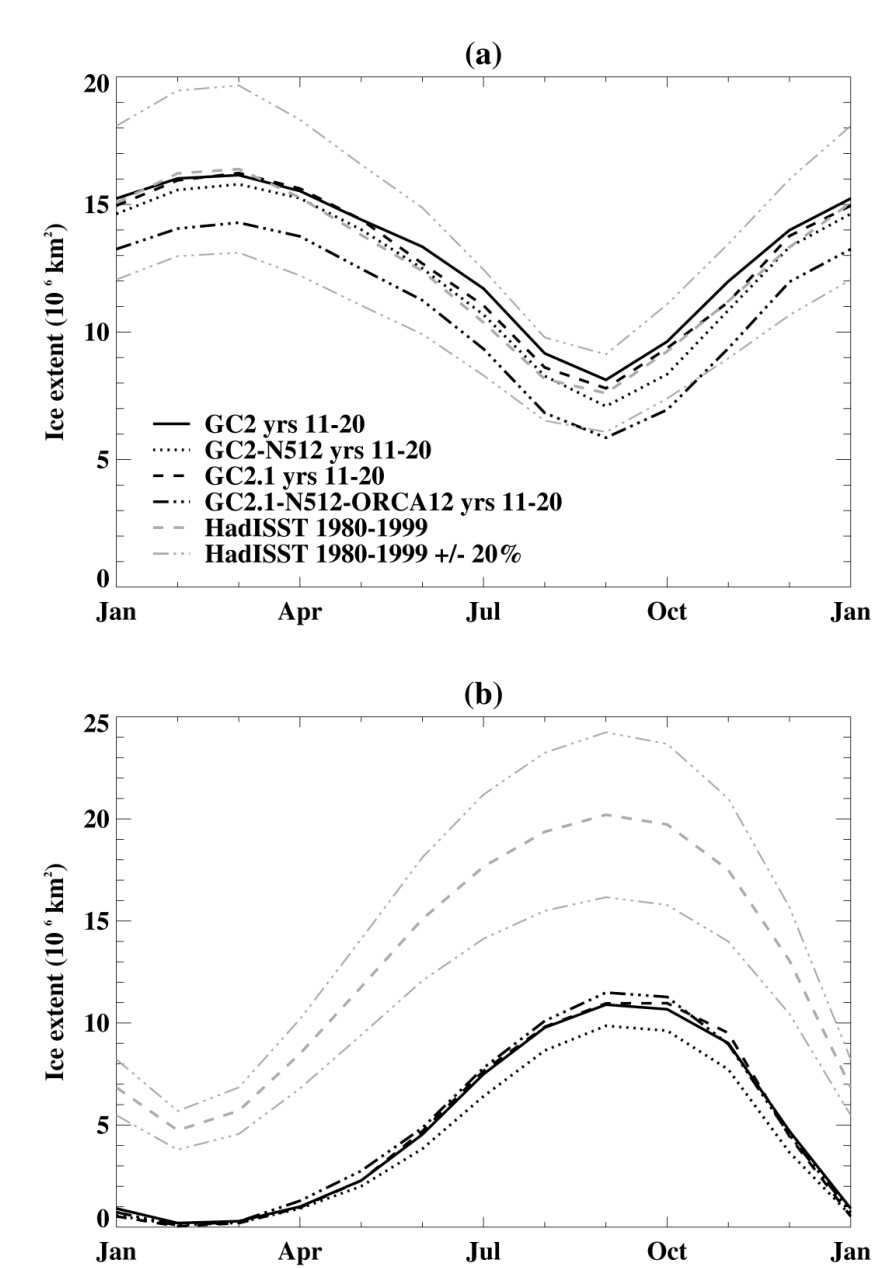

Figure 5: Seasonal cycle of sea ice extent in a) Northern and b) Southern hemisphere for years
11-20 compared against HadISST 1980-99 and with +/- 20% error bars denoted.



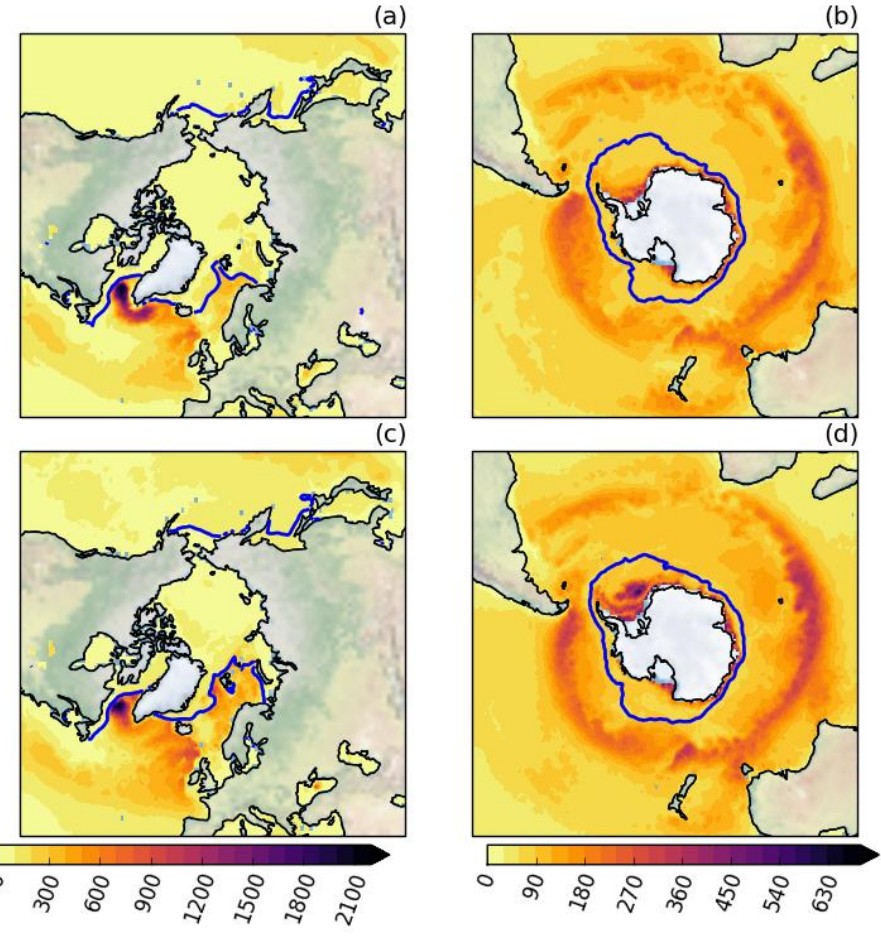

Figure 6: Mean March Northern hemisphere winter mixed layer depth (m) and Arctic sea ice
edge and mean September Southern hemisphere winter mixed layer depth (m) and sea ice
edge for years 11-20 for GC2 (a,b) and GC2.1-N512O12 (c,d)





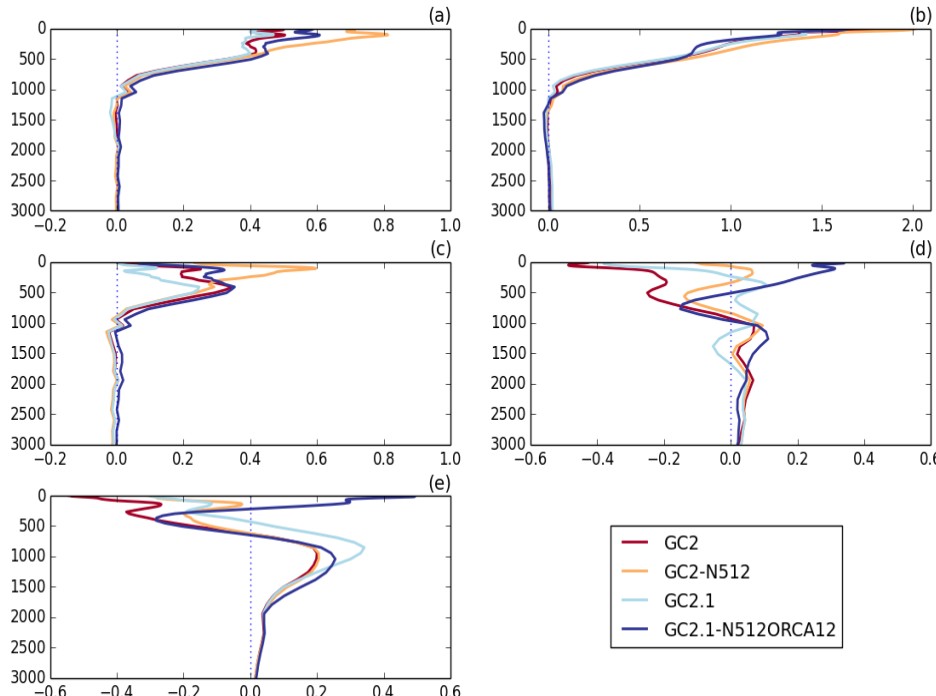

Figure 7: Area-weighted mean temperature difference (years 11-20 minus year 1; °C) for
GC2, GC2-N512, GC2.1 and GC2.1-N512O12 for a) global, b) 90S-30S, c) 30S-30N, d)
30N-90N, e) 65N-90N. Note the range on the x-axis is equal in all panels except (b). The
vertical axis denotes depth (m).





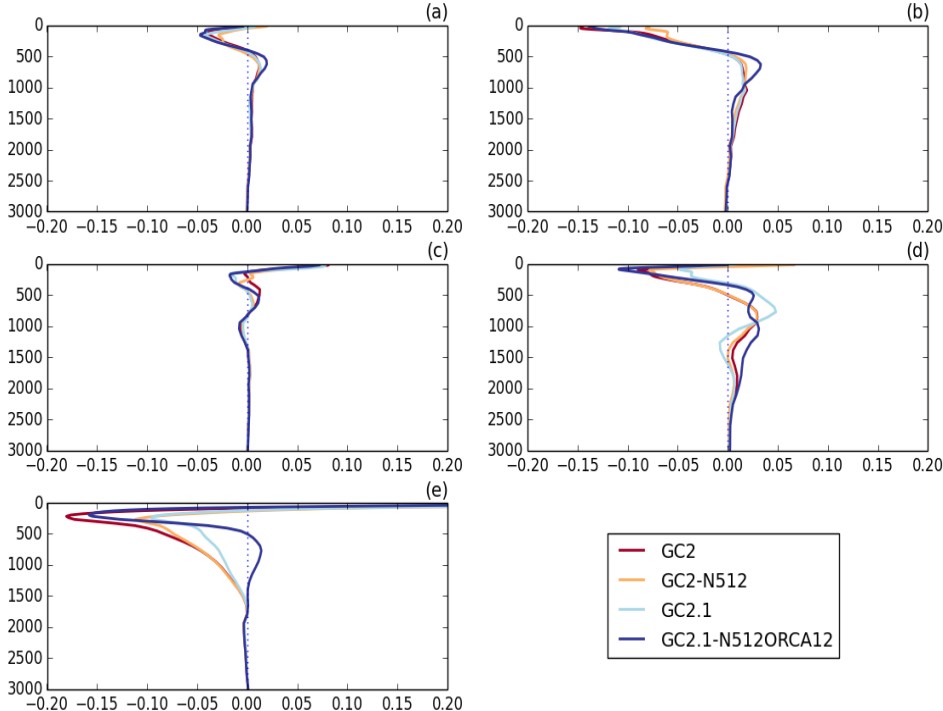

2    Figure 8: Area-weighted mean salinity difference (years 11-20 minus year 1; psu) for GC2,

3    GC2-N512, GC2.1 and GC2.1-N512O12 for a) global, b) 90S-30S, c) 30S-30N, d) 30N-90N,

4    e) 65N-90N. The vertical axis denotes depth (m).



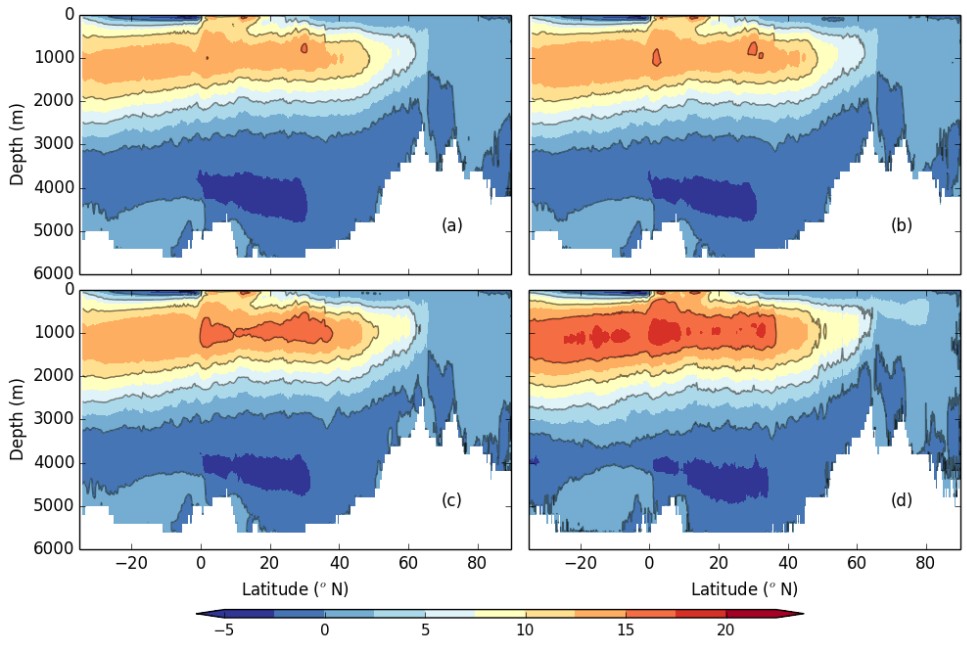

4    Figure 9: Atlantic Meridional overturning for (a) GC2, (b) GC2-N512, (c) GC2.1 and (d)

5    GC2.1-N512O12, meaned over years 11-20. Contours in Sverdrups ($10^6$ m$^3$s$^{-1}$), with line

6    contour spacing of 5 Sv.



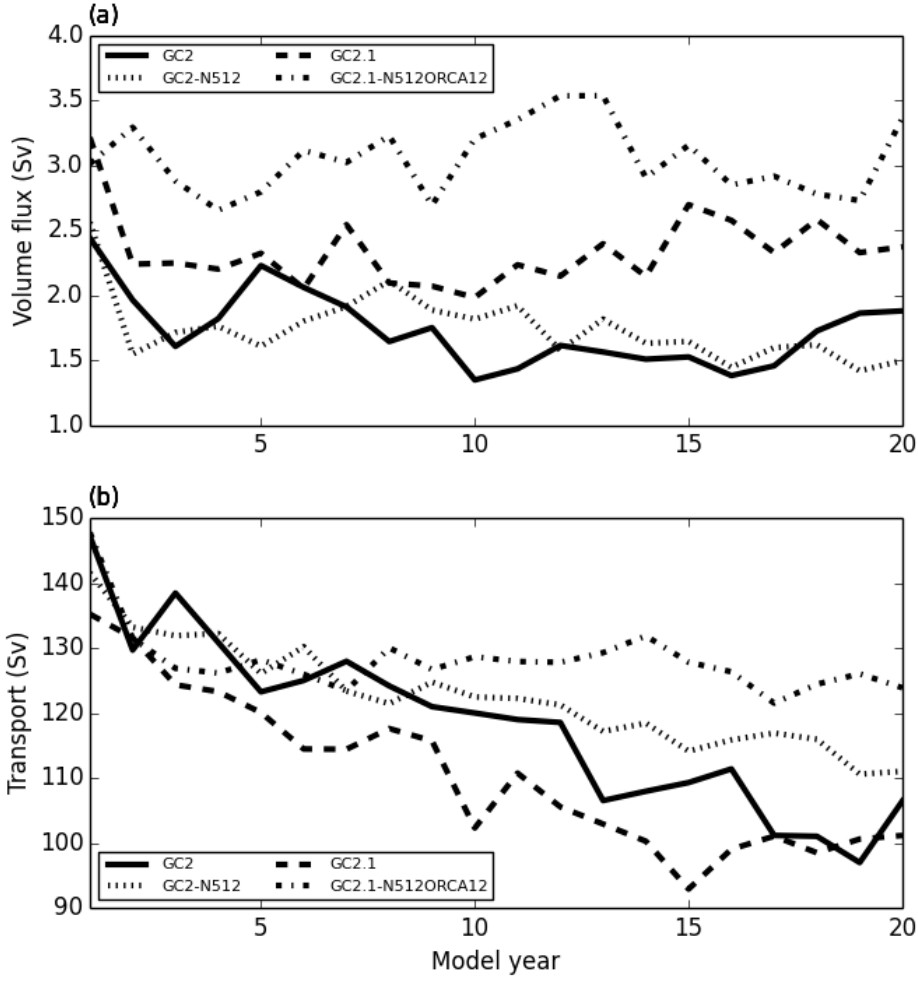

4    Figure 10: a) Denmark Straits volume flux (Sv) (calculated as southward flow across the

5    Greenland-Iceland-Scotland ridge below density of 27.8 kg m$^{-3}$) and b) Antarctic Circumpolar

6    Current transport (Sv) from GC2, GC2-N512, GC2.1 and GC2.1-N512O12





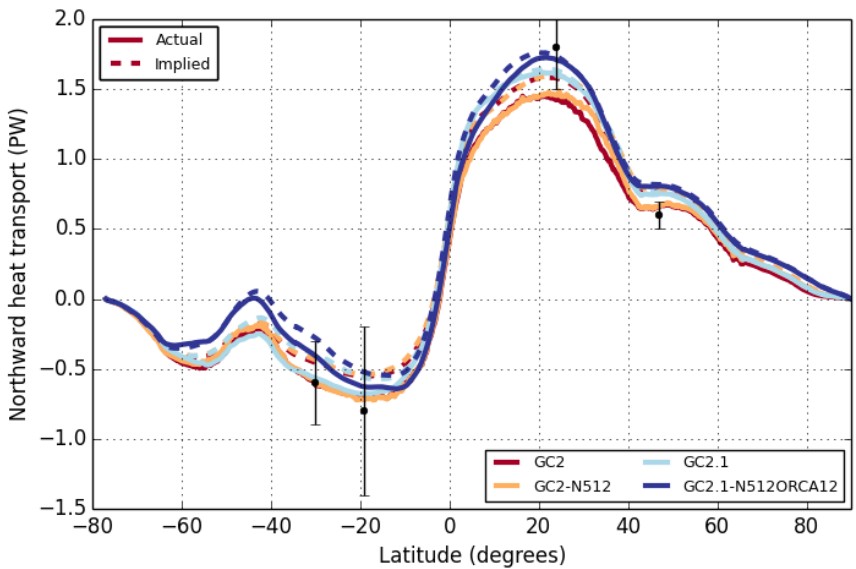

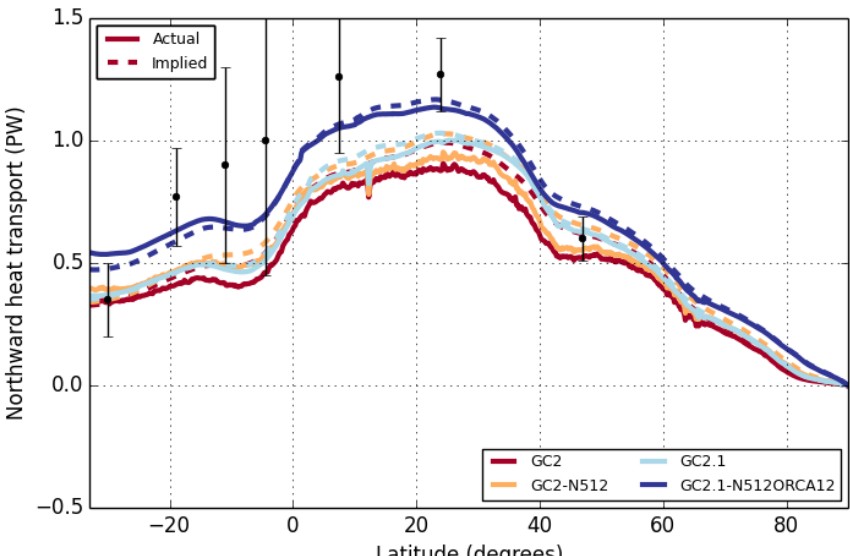

Figure 11: Actual (bold) and implied (dashed) northward heat transports from GC2, GC2-
N512, GC2.1 and GC2.1-N512O12 for (a) global and (b) Atlantic basins. The implied
transport (integrated southwards from the pole using the ocean surface heat flux) uses heat
fluxes in which the global mean imbalance has been removed at every point. Observational
estimates and associated error bars from Ganachaud and Wunsch (2003) are shown.