# Peer review of "The impact of resolving the Rossby radius at mid-latitudes in the ocean: results from a high-resolution version of the"

_Geoscientific Model Development, 2016_

## Referee Comment (RC1) · S. M. Griffies (Referee) · 29 Apr 2016

Review from Stephen Griffies, NOAA/GFDL

This is generally a well written and concise entree into a suite of coupled climate models run for 20 years. To my knowledge the finest resolution model, GC2.1-N512O12, is the state-of-the-science, at least for global models run for more than a few years. This point is worth emphasizing.

The tasks required are immense to produce a sensible simulation, even for the rather brief 20 years considered here. I applaud this effort, though note it is far from complete!

Yet as an introduction to the model suite, this is a useful contribution to the literature, and it provides an important peer-reviewed touchpoint for the developers.

This manuscript is appropriate for GMD. I recommend publication after minor revisions.

General points/questions:

–Others who have developed models of this resolution with refined coupling periods (hourly or smaller) sometimes have problems related to coupled ocean/sea ice instabilities as discussed by Hallberg (CLIVAR Exchanges, No.65 (Vol 19 No.2) July 2014). It would serve the reader to know if you encountered any similar instabilities, and if so, what methods were used to suppress them. If you did not encounter such instabilities, it would be useful to state that as well.

–The Weddell Sea polynya in GC2.1-N512O12 warrants more discussion. In similar models at GFDL, we have seen that such polynyas can increase ACC transport, much as noted on page 10, lines 10-22. Do any of the other models in your suite have a polynya? Is the polynya large in area and going very deep? How long does it last? I am puzzled that the SST biases in Figures 2 and 3 show no sign of the polynya. In other models I have seen, such polynyas increase SST due to release of mid-depth heat. That SST signature is missing here. Perhaps the polynya is only for a year or two, and is averaged out by the 10 year mean? Please discuss, as this is an important feature to expose.

Minor points.

pg1,line18: I appreciate that it is the surface ocean that the atmosphere cares about, and the sentence is referring to air-sea fluxes. But the sentence can be construed, incorrectly, to mean that ONLY surface eddies and boundary currents are necessary to do air-sea coupling right. As the authors show in this paper, there is more to air-sea fluxes than the surface ocean. For example, overflows and the AMOC are key. So I recommend finding a different way to write this sentence.

pg1,line23: Admittedly a picky point, but worth being precise: hours are listed here as "frequency" for coupling (1-hour versus 3-hour). In fact, these are the "coupling periods" not the "coupling frequencies".

pg3, line5: Behrens et al. (2013) should be Behrens (2013). This citation refers to a single-authored PhD thesis.

pg4,line9: again, "3-hourly to hourly" refers to "coupling period" not "coupling frequency".

pg4, line 11: it is not clear what model is being referred to here when discussing the time step. I assume the ocean, but it should be clearly stated.

pg4, line 27: Viscosity is a positive number. The biharmonic operator carries the negative sign. Please change. Doing so will also make the sentence correct. Namely, it presently reads "a reduction in the bilaplacian viscosity from -5e11 to -0.25e10". With the minus sign, this is not a reduction, but an increase! Again, just drop the minus signs on the viscosity so that all will make sense.

pg5, line 5: Including tides generally increases the flow speed in simulations. So what you mean here is that there is missing "tidal dissipation" in the model. That is, you are not suffering from missing tides, but instead suffereing from missing tidal dissipation.

pg5,line6: what is "atmospheric theta"? Please define the jargon.

pg5, line 28: what sort of "instabilities" do you find enhanced with the finer atmosphere? Those instabilities discussed earlier near the UK due to missing tidal dissipation? Something else?

pg8,lines10-11: More heat into the ocean interior is NOT what Griffies et al (2015) found with the GFDL CM2.6 simulation (1/10th degree ocean) relative to the coarser ocean (1/4th degree) in CM2.5. Instead, enhanced mesoscale eddy actitivity led to less heat entering the ocean. So...why does GC2.1-N512O12 get warmer in the interior than GC2-N512? Could it be an increase in spurious diapycnal diffusion from advection

errors? It is useful to speculate here, even if you do not perform a budget analysis as in Griffies et al.

pg9, It is useful to state how the mixed layer depth is computed.

pg10, line6: a more recent Denmark St overflow measurement paper is Jochumsen et al. (2012), 10.1029/2012JC008244

pg10,line18: a more recent Drake Passage transport paper is Meredith et al. (2011) 10.1029/2010RG000348

pg10, equation (1): dS is not defined. I presume it is dS = dx * dz. Please specify.

pg10,line31: A_iso is not the "isopycnal diffusion". Instead, it is the "isopycnal diffusivity". Or more properly, it is the "isoneutral diffusivity", which is consistent with terminology used elsewhere in this manuscript.

pg11, line1-3: I puzzled by this discussion. You state that the isoneutral fluxes are smaller than other terms, but then state, parenthetically, that the dianeutral diffusive fluxes are very small when integrated over the full depth. I think there is some confusion here.

In particular, the dianeutral diffusive fluxes, which are computed as vertical diffusion in NEMO, should have a zero depth integral since vertical diffusion only redistributes heat within a column. In contrast, the depth integrated isoneutral diffusion fluxes have a nonzero depth integral.

Are you arguing that the depth integrated isoneutral diffusive heat fluxes are small?

pg11, lines23-24: I fail to see how removing a global mean from the right hand side of equation (1) will not be seen by the left hand side of equation (1). Is that what you want? Please detail more of what you mean by "subtracting the global mean imbalance from the surface fluxes before integrating zonally and meridionally." It is vague as stated.

pg12, line13-ff: Again, I wonder how much of what you are seeing relates to the Wed-

dell Sea polynya.

pg13,line32: GFDL prediction folks claim that eddying oceans are too expensive for initialization schemes. So they are not pursuing ocean resolution. That contrasts with your motivation at the Met Office. The community would be well served to know more about your initialization strategies with an eddying ocean. It is worth at least a paragraph.

Figure 1: legend font is tiny; needs to be larger.

Figures 2,3,4: is the land/sea mask based on the model grid, or based on an observed topography dataset? It looks like observed. I suggest it more useful and honest for a modelling paper to show the land/sea mask based on the model grid.

I also dislike white land since there is also a white part of the colour bar for ocean fields. I suggest colouring the land light brown or light gray, in order to clearly distinguish water from land.

Figure 3: the colour range should be smaller in order to better see the anomalies.

Figure 5: the HadISST values should be coloured to better distinguish from the many model lines. The present light gray shading does not come through well.

Figure 6: how deep does the MLD penetrate in the saturated regions? This issue goes to the question about how significant is the polynya.

Figure 9: The bottom topography appears nearly the same across the ocean resolutions. Are you sure you are showing the proper bottom?

Does the model make use of the partial bottom cells? If so, then the bottom shown here does not appear to reflect the partial cells; this instead figure looks like it is showing full cells. Again, it is preferable to show the what the model is actually using.

END OF REVIEW

---

## Referee Comment (RC2) · A. Hogg (Referee) · 21 May 2016

This manuscript describes the development of a version of the UKMO GC2 coupled climate model with enhanced resolution in both atmosphere and ocean, as well increased coupling frequency. The development of this model is a significant achievement - and at 1/12° ocean resolution is, to my knowledge the highest resolution coupled model available. In addition, and contrary to many previous coupled models with high ocean resolution, the authors systematically include the effect of enhanced atmospheric resolution.

However, the technical achievements outlined here are not quite matched by the depth of analysis of the model results. In many cases, changes between results from different simulations are causally attributed with only a superficial analysis. I accept that, for GMD readers, the attribution of different physical effects may be of secondary importance to the technical achievement; but if the authors want to imply causality then a more rigorous analysis is required. In a number of cases (details below) the authors could sidestep this issue by rephrasing the text - i.e., by making it clear that they are speculating on the cause rather than attributing, and pointing out where additional experiments will enable them to resolve the uncertainty. If these issues are addressed I would be happy to recommend this paper as a suitable contribution to GMD.

Major comments

On p.4 (line 26) it is noted that the transition from ORCA025 to ORCA12 is accompanied by a reduction in the isoneutral diffusivity from 300 to 125 mˆ2/s. It would help to have a justification of this change - in particular, if eddies are fully resolved, why do we need isoneutral diffusivity at all? If it is needed, then on what basis do we choose 125? This question is relevant because, for example, the reduction in SST biases is attributed to resolution (p. 6, line 24). However, this result (at least for the Southern Ocean warm bias) might alternatively be be attributed to reduced parameterised upwards eddy heat flux. This effect may be consistent with the analysis on p.11, which shows a reduction in the time-mean southward heat transport at southern latitudes.And, finally, in the discussion there is reference to previous experiments in which changes in isoneutral diffusivity are associated with high-latitude cooling, but the authors argue that this "is believed to be a secondary effect" due to the long timescales associated with that paper.

This is one example where the authors need to pose one of two possible causes (with further experiments to tease out the root cause) or else perform a more in-depth analysis. For this case there are some clear, but simple, tests which could be performed. The isoneutral diffusivity contribution to the southward (or upwards) heat flux could be

calculated explicitly. Alternatively, this question could be resolved by one additional GC2.1 simulation with reduced isoneutral diffusivity.

I wasn't entirely convinced by the description of the MOC changes (bottom of p. 9). Firstly, it is argued that changes are dominated by the cell associated with NADW - this may be true, but the other cells are not shown. This manuscript would be much more complete if the full MOC were shown, including the Southern Ocean (which would require transforming the overturning analysis into density space). In addition, the attribution of both NADW formation and Denmark Strait outflow increases to higher resolution seems fraught; the GC2.1 case sees a modest increase in both of these quantities, implying that the higher coupling frequency is partly response for the changes.

On p.10, l. 17, the ACC transport increase at higher resolution is noted as being consistent with both enhanced NADW and the Weddell Sea polynya. It seems unlikely that NADW formation can affect ACC transport in a short 20 year run (see Allison et al., JMR, 2011) - meaning that it is most likely that the Weddell Sea effect is dominating. Either way, both effects probably need to be supported in the form of a reference to existing literature. On a similar note, it seems likely that the small ACC transport in these simulations may be linked to weak AABW formation because of the Southern Ocean SST bias. This point could be further clarified if the full MOC were shown as suggested above.

The paragraph starting on p. 12, l. 29, is somewhat unconvincing. The case is made that refining both atmosphere and ocean resolution is important to gain the full benefit of resolution improvements. Yet, for almost all the metrics shown here, the N512 case showed only minor differences from GC2 (as noted in the first paragraph in this section). It may be that there are other metrics on which the N512 case performs well, but they are not shown here, so should not be included in the summary of this paper.

Minor points: - p. 4, line 22: I'm not sure I would call this an aspect ratio. Maybe just ratio?

- p. 6, line 9: It seems to me that the Southern Ocean SST biases here are larger than they were for CMIP-5. If true, then this should be explicitly stated, along with a reference to the published bias (it looks as if you're hiding something by stressing the pattern, rather than magnitude, of the bias).

- Several times through the manuscript the N512O12 simulation is listed as N512-ORCA12 - best to be consistent if possible. (p. 6, l.24; p. 7, l.24; legend of Fig. 5, )

- p.10, l.31: isopyncnal -> isopycnal

- p. 11, l.4: There are four instances of "change/s" in the one sentence here, which becomes a little repetitive.

- p. 12, l.17: I'm not convinced that we expect more slumping of ACC isopycnals in the eddy-resolving simulation - changes in eddy KE are more likely to control the ACC through enhanced vertical momentum transport - but if there is a previously published expectation supporting this statement then I suggest a reference.

- The reference to "seamless" prediction makes little sense to those outside the UKMO community, and I suggest it should be either explained to great depth, or removed.

- p. 35, l.6: specify "north pole".

---

## Author Comment (AC1) · 9 Jul 2016

Review by Stephen Griffies

We thank Dr. Griffies for his constructive comments and address them here. References refer to the manuscript with tracked changes.

*This is generally a well written and concise entree into a suite of coupled climate models run for 20 years. To my knowledge the finest resolution model, GC2.1-N512O12, is the state-of-the-science, at least for global models run for more than a few years. This point is worth emphasizing.*

- A sentence has been included (p3, line1) to emphasise that with the resolution of atmosphere and ocean components and hourly coupling, this model is state of the art.

*The tasks required are immense to produce a sensible simulation, even for the rather brief 20 years considered here. I applaud this effort, though note it is far from complete!*

*Yet as an introduction to the model suite, this is a useful contribution to the literature, and it provides an important peer-reviewed touchpoint for the developers.*
*This manuscript is appropriate for GMD. I recommend publication after minor revisions.*

*Others who have developed models of this resolution with refined coupling periods (hourly or smaller) sometimes have problems related to coupled ocean/sea ice instabilities as discussed by Hallberg (CLIVAR Exchanges, No.65 (Vol 19 No.2) July 2014).*
*It would serve the reader to know if you encountered any similar instabilities, and if so, what methods were used to suppress them. If you did not encounter such instabilities, it would be useful to state that as well.*

- We did not experience instabilities associated with the hourly coupling. This is stated (p4, lines 24-25)

*The Weddell Sea polynya in GC2.1-N512O12 warrants more discussion. In similar models at GFDL, we have seen that such polynyas can increase ACC transport, much as noted on page 10, lines 10-22. Do any of the other models in your suite have a polynya? Is the polynya large in area and going very deep? How long does it last? I am puzzled that the SST biases in Figures 2 and 3 show no sign of the polynya. In other models I have seen, such polynyas increase SST due to release of mid-depth heat. That SST signature is missing here. Perhaps the polynya is only for a year or two, and is averaged out by the 10 year mean? Please discuss, as this is an important feature to expose.*

- Further investigation shows that the polynya first appears in year 12 and varies from year-to-year. In years without a polynya, the maximum mixed layer depth is less than 1000m. However, in year 15, the maximum mixed layer depth is 1556 m and in year 20 it is 2070m. More information about the polynya and its variability is given on p8, line 7, p10, lines 6-10 and in the new figure 9.

*Pg1, line18: I appreciate that it is the surface ocean that the atmosphere cares about, and the sentence is referring to air-sea fluxes. But the sentence can be construed, incorrectly, to mean that ONLY surface eddies and boundary currents are necessary to do air-sea coupling right. As the authors show in this paper, there is more to air-sea*

*fluxes than the surface ocean. For example, overflows and the AMOC are key. So I recommend finding a different way to write this sentence.*

- Sentence has been changed in abstract (p1, line 17-18)

*Pg1,line23: Admittedly a picky point, but worth being precise: hours are listed here as "frequency" for coupling (1-hour versus 3-hour). In fact, these are the "coupling periods" not the "coupling frequencies".*

- P1, line 23 frequency changed to period (also p4, line 14 and in table 1)

*Pg3,line5: Behrens et al. (2013) should be Behrens (2013). This citation refers to a single-authored PhD thesis.*

- P3, line 8 reference corrected

*Pg4,line9: again, "3-hourly to hourly" refers to "coupling period" not "coupling frequency*
- P4, line 14 corrected to period

*Pg4,line11: it is not clear what model is being referred to here when discussing the time step. I assume the ocean, but it should be clearly stated.*
- P4, line 16 ocean inserted

*Pg4,line27: Viscosity is a positive number. The biharmonic operator carries the negative sign. Please change. Doing so will also make the sentence correct. Namely, it presently reads "a reduction in the bilaplacian viscosity from -5e11 to -0.25e10". With the minus sign, this is not a reduction, but an increase! Again, just drop the minus signs on the viscosity so that all will make sense.*
- P5, line 4 viscosity sign corrected

*Pg5,line5: Including tides generally increases the flow speed in simulations. So what you mean here is that there is missing "tidal dissipation" in the model. That is, you are not suffering from missing tides, but instead suffereing from missing tidal dissipation.*
- P5, line 15 tides changed to tidal dissipation

*Pg5,line6: what is "atmospheric theta"? Please define the jargon.*
- P5, line 17 theta changed to temperature.

*Pg5,line28: what sort of "instabilities" do you find enhanced with the finer atmosphere? Those instabilities discussed earlier near the UK due to missing tidal dissipation? Something else?*
- P6, lines 7- 8 clarification added to the instabilities

*Pg8,lines10-11: More heat into the ocean interior is NOT what Griffies et al (2015) found with the GFDL CM2.6 simulation (1/10th degree ocean) relative to the coarser ocean (1/4th degree) in CM2.5. Instead, enhanced mesoscale eddy actitivity led to less heat entering the ocean. So...why does GC2.1-N512O12 get warmer in the interior*

*than GC2-N512? Could it be an increase in spurious diapycnal diffusion from advection errors? It is useful to speculate here, even if you do not perform a budget analysis as in Griffies et al.*

- Paragraph has been inserted p8, lines 26-32 to discuss the heat uptake issue and contrast with Griffies et al. (2015)

*Pg9, It is useful to state how the mixed layer depth is computed.*

- This has been added to a footnote on p9

*Pg 10,line6: a more recent Denmark St overflow measurement paper is Jochumsen et al. (2012), 10.1029/2012JC008244*

- P11, line 4 Jochumsen reference added

*pg10,line18: a more recent Drake Passage transport paper is Meredith et al. (2011) 10.1029/2010RG000348*

- P11, line 13 Meredith reference added

*Pg10, equation (1): dS is not defined. I presume it is dS = dx \* dz. Please specify.*
- P12, lines 4-6 definition of dS and also dA added

*pg10,line31: A_iso is not the "isopycnal diffusion". Instead, it is the "isopycnal diffusivity". Or more properly, it is the "isoneutral diffusivity", which is consistent with terminology used elsewhere in this manuscript.*
- P12 line 3 changed to isoneutral diffusivity and on p14, lines 25-26

*pg11, line1-3: I puzzled by this discussion. You state that the isoneutral fluxes are smaller than other terms, but then state, parenthetically, that the dianeutral diffusive fluxes are very small when integrated over the full depth. I think there is some confusion here.*
*In particular, the dianeutral diffusive fluxes, which are computed as vertical diffusion in NEMO, should have a zero depth integral since vertical diffusion only redistributes heat within a column. In contrast, the depth integrated isoneutral diffusion fluxes have a nonzero depth integral.*
*Are you arguing that the depth integrated isoneutral diffusive heat fluxes are small?*

- P12 line 7 This comment is correct. The wording has changed to isoneutral

*pg11, lines23-24: I fail to see how removing a global mean from the right hand side of equation (1) will not be seen by the left hand side of equation (1). Is that what you want? Please detail more of what you mean by "subtracting the global mean imbalance from the surface fluxes before integrating zonally and meridionally." It is vague as stated.*
- Removing the global mean surface flux from the rhs of (1) is equivalent to removing the global integral of the temperature drift from the lhs of (1). This means that the imbalance represents the residual local drifts. Text has been added to make this clear (p12, line 31 – p13, line 4)

*pg12, line13-ff: Again, I wonder how much of what you are seeing relates to the Weddell Sea polynya*

- A caveat has been added on p13, line 31-p14, line 1

*pg13,line32: GFDL prediction folks claim that eddying oceans are too expensive for initialization schemes. So they are not pursuing ocean resolution. That contrasts with your motivation at the Met Office. The community would be well served to know more about your initialization strategies with an eddying ocean. It is worth at least a paragraph.*

- A short paragraph on initialisation has been included on p15 lines 26- p16, line 2

*Figure 1: legend font is tiny; needs to be larger.*

- Figure 1: legend size increased

*Figures 2,3,4: is the land/sea mask based on the model grid, or based on an observed topography dataset? It looks like observed. I suggest it more useful and honest for a modelling paper to show the land/sea mask based on the model grid.*
*I also dislike white land since there is also a white part of the colour bar for ocean fields. I suggest colouring the land light brown or light gray, in order to clearly distinguish water from land.*

- Figures 2,3,4: land has been changed to grey and mask from model grid used

*Figure 3: the colour range should be smaller in order to better see the anomalies.*

- Figure 3: range reduced to -2 to 2

*Figure 5: the HadISST values should be coloured to better distinguish from the many model lines. The present light gray shading does not come through well.*

- Figure 5: HadISST values coloured. Legend changed to GC2.1-N512O12

*Figure 6: how deep does the MLD penetrate in the saturated regions? This issue goes to the question about how significant is the polynya.*

- Figure 6: Maximum MLD in the average is 788m. However, as seen in the extra figure, the polynya only appears in the last 9 years of the simulation and then it is only in the final year that it extends down to below 2000m. Land has been changed to grey and definition of sea ice edge added to caption.

*Figure 9: The bottom topography appears nearly the same across the ocean resolutions. Are you sure you are showing the proper bottom?*
*Does the model make use of the partial bottom cells? If so, then the bottom shown here does not appear to reflect the partial cells; this instead figure looks like it is showing full cells. Again, it is preferable to show the what the model is actually using.*

- Figure 9 (now figure 10): The model does use partial cells and the average topography is changing between ORCA025 and ORCA12

---

## Author Comment (AC2) · 9 Jul 2016

We thank Dr Hogg for his constructive comments and address them here. References refer to the manuscript with tracked changes.

*This manuscript describes the development of a version of the UKMO GC2 coupled climate model with enhanced resolution in both atmosphere and ocean, as well increased coupling frequency. The development of this model is a significant achievement - and at 1/12_ ocean resolution is, to my knowledge the highest resolution coupled model available. In addition, and contrary to many previous coupled models with high ocean resolution, the authors systematically include the effect of enhanced atmospheric resolution.*

*However, the technical achievements outlined here are not quite matched by the depth of analysis of the model results. In many cases, changes between results from different simulations are causally attributed with only a superficial analysis. I accept that, for GMD readers, the attribution of different physical effects may be of secondary importance to the technical achievement; but if the authors want to imply causality then a more rigorous analysis is required. In a number of cases (details below) the authors could sidestep this issue by rephrasing the text - i.e., by making it clear that they are speculating on the cause rather than attributing, and pointing out where additional experiments will enable them to resolve the uncertainty. If these issues are addressed I would be happy to recommend this paper as a suitable contribution to GMD.*

We have made a number of revisions which we hope address the comments.

*On p.4 (line 26) it is noted that the transition from ORCA025 to ORCA12 is accompanied by a reduction in the isoneutral diffusivity from 300 to 125 m^2/s. It would help to have a justification of this change - in particular, if eddies are fully resolved, why do we need isoneutral diffusivity at all? If it is needed, then on what basis do we choose 125? This question is relevant because, for example, the reduction in SST biases is attributed to resolution (p. 6, line 24). However, this result (at least for the Southern Ocean warm bias) might alternatively be be attributed to reduced parameterised upwards eddy heat flux. This effect may be consistent with the analysis on p.11, which shows a reduction in the time-mean southward heat transport at southern latitudes.And, finally, in the discussion there is reference to previous experiments in which changes in isoneutral diffusivity are associated with high-latitude cooling, but the authors argue that this "is believed to be a secondary effect" due to the long timescales associated with that paper. This is one example where the authors need to pose one of two possible causes (with further experiments to tease out the root cause) or else perform a more in-depth analysis. For this case there are some clear, but simple, tests which could be performed.The isoneutral diffusivity contribution to the southward (or upwards) heat flux could be calculated explicitly. Alternatively, this question could be resolved by one additional GC2.1 simulation with reduced isoneutral diffusivity.*

- We agree that the isoneutral diffusivity needs more discussion.
    - We include the following text on p5, lines 7-11 'While reducing the isoneutral tracer diffusivity is consistent with the increase in resolution, we note that results may have some sensitivity to its magnitude. Experiments to investigate the impact of this parameter in GC2 were not performed but will be pursued in future work with GC3 (the next version of the coupled model).'

    o   In the discussion, the following text has been included on p14, lines 28-32 'Given that the results here exhibit some consistency with those of Pradal and Gnanadesikan (2014) in the Southern Ocean, further work is required to quantify the role of isoneutral diffusivity in producing changes in SST on decadal timescales.'

*I wasn't entirely convinced by the description of the MOC changes (bottom of p. 9). Firstly, it is argued that changes are dominated by the cell associated with NADW - this may be true, but the other cells are not shown. This manuscript would be much more complete if the full MOC were shown, including the Southern Ocean (which would require transforming the overturning analysis into density space). In addition, the attribution of both NADW formation and Denmark Strait outflow increases to higher resolution seems fraught; the GC2.1 case sees a modest increase in both of these quantities, implying that the higher coupling frequency is partly response for the changes.*

- Unfortunately, we did not have 5 day means of the full velocity field saved to allow calculation of the overturning in density space. As many authors have shown, this field is not entirely meaningful unless the eddy component is included. Calculations of the overturning in density space with monthly means suggest that the changes in NADW due to resolution and coupling period are robust but we can't be certain about the AABW cell. The 5 day mean velocity fields was an oversight in setting up the model diagnostics and we will address this issue in future runs with the GC3 model for CMIP6. A comment has been added to the paper on this point (p10, lines 28-30).
- We have however modified the text on p10, lines 21-26 to reflect that the coupling frequency plays a role as well as on p11, line 6 and p14, lines 12-13

*On p.10, l. 17, the ACC transport increase at higher resolution is noted as being consistent with both enhanced NADW and the Weddell Sea polynya. It seems unlikely that NADW formation can affect ACC transport in a short 20 year run (see Allison et al., JMR, 2011) - meaning that it is most likely that the Weddell Sea effect is dominating. Either way, both effects probably need to be supported in the form of a reference to existing literature. On a similar note, it seems likely that the small ACC transport in these simulations may be linked to weak AABW formation because of the Southern Ocean SST bias. This point could be further clarified if the full MOC were shown as suggested above.*

- P11, lines 20-23 we have included a reference to Jones et al. (2011) who show that transient responses in the ACC can be seen within 10 years in their idealised experiments.
- References have been added for the NADW and the Weddell Sea links to the ACC (p11, lines 17-18)

*The paragraph starting on p. 12, l. 29, is somewhat unconvincing. The case is made that refining both atmosphere and ocean resolution is important to gain the full benefit of resolution improvements. Yet, for almost all the metrics shown here, the N512 case*

*showed only minor differences from GC2 (as noted in the first paragraph in this section). It may be that there are other metrics on which the N512 case performs well, but they are not shown here, so should not be included in the summary of this paper.*

- The paragraph p15, lines 11-17 has been changed in response to the comments to state that ocean resolution and coupling frequency is important and states that further work is required to quantify whether a high resolution atmosphere component is required.

*p. 4, line 22: I'm not sure I would call this an aspect ratio. Maybe just ratio?*
- P4, line 29 aspect deleted

*p. 6, line 9: It seems to me that the Southern Ocean SST biases here are larger than they were for CMIP-5. If true, then this should be explicitly stated, along with a reference to the published bias (it looks as if you're hiding something by stressing the pattern, rather than magnitude, of the bias).*
- Comments on the magnitude of SST biases in GC2 have been added on p6 lines 21-25. Basically the SST warmed everywhere in GC2 relative to the CMIP5 model HadGEM2-AO (see figure 1 of Williams et al., 2015) leading to improvements in the Northern hemisphere and degradations in the Southern hemisphere.

*Several times through the manuscript the N512O12 simulation is listed as N512-ORCA12 - best to be consistent if possible. (p. 6, l.24; p. 7, l.24; legend of Fig. 5)*
- Corrected to GC2.1-N512O12 in text and figures 5, 7, 8, 11, 12

*p.10, l.31: isopyncnal -> isopycnal*
- Changed isopycnal to isoneutral p12, line 3

*p. 11, l.4: There are four instances of "change/s" in the one sentence here, which becomes a little repetitive.*
- Text on p12, lines 18-22 changed to improve reading

*p. 12, l.17: I'm not convinced that we expect more slumping of ACC isopycnals in the eddy-resolving simulation - changes in eddy KE are more likely to control the ACC through enhanced vertical momentum transport - but if there is a previously published expectation supporting this statement then I suggest a reference.*
- Text has been changed on p13, lines 27- p14, line 1 to reflect the discussion on the ACC

*The reference to "seamless" prediction makes little sense to those outside the UKMO community, and I suggest it should be either explained to great depth, or removed.*
- P15, line 18 seamless removed

*p. 35, l.6: specify "north pole".*

- P49. North Pole specified

---

## Author Response (AR2)

Response to Andy Hogg

Many thanks for your comments. We have made the following changes:

-The ACC transport section has been modified to address the comments. As discussed in the
revised text, approximately two-thirds of the density gradient difference between GC2.1-
N512O12 and GC2 is due to denser water to the south of the ACC in GC2.1-N512O12
consistent with the polynya formation. The reference to Jones has been removed.

-The Southern Ocean SST bias approximately doubles between HadGEM2 and GC2. The
causes of the change in the bias will be described in a paper led by Pat Hyder which is in
preparation.

We totally agree regarding the overturning in density space and we are working on including
this diagnostic in our future runs.

[revised manuscript text omitted]